# Are consumer confidence and asset value expectations positively associated with length of daylight?: An exploration of psychological mediators between length of daylight and seasonal asset price transitions

**Yoichi Sekizawa**[ID]◉*, **Yoko Konishi**◉

Research Institute of Economy, Trade and Industry, Tokyo, Japan

◉ These authors contributed equally to this work.
* sekizawa-yoichi@rieti.go.jp

**Data Availability Statement:** Data used in this study (the Consumer Confidence Survey of Japan) are owned by a third-party organization (the

## Abstract

Many economists claim that asset price transitions, particularly stock price transitions, have a seasonal cycle affected by length of daylight. Although they claim that the seasonal affective disorder (SAD) is a mediator between the length of daylight and asset price transitions, recent studies in psychology have been inconclusive about the existence of SAD, and some economics studies disagree regarding the involvement of SAD in seasonal stock price transitions. The purpose of the present study is to examine if there is any psychological mediator linking length of daylight and seasonal asset price transitions as an alternative or supplement to SAD. As a possible mediator, we examined Japan's consumer confidence index (CCI) and asset value expectations (AVE), which indicate people's optimism for future economy and are generated from a monthly household survey by the Japanese government. We analyzed individual longitudinal data from this survey between 2004 and 2018 and estimated four fixed-effects regression models to control for time-invariant unobserved heterogeneity across individual households. The results revealed that, (i) there was a seasonal cycle of CCI and AVE; the trough occurred in December and the peak in early summer; (ii) the length of daylight time was positively associated with CCI and AVE; and (iii) the higher the latitude, the larger the seasonal cycle of CCI and AVE became. These findings suggest that the length of the daylight may affect asset price transitions through the cycle of optimism/pessimism for future economy exemplified by the CCI and AVE.

## Introduction

Many studies in economics have shown that asset price transitions, particularly stock price transitions, have a seasonal cycle [1–13]. Although the causes of this cycle have not been sufficiently identified, several studies attributed this seasonal cycle to the length of daylight [2–8, 13]. They further hypothesized that the seasonal emotional cycle exemplified by the seasonal affective disorder (SAD) [14] is a mediator linking the asset price transition and the length of

Cabinet Office of the Japanese Government). They are accessible via a request to the Cabinet Office based on Article 33 of the Statistics Act. The initial contact address for the data request is Economic and Social Research Institute (ESRI), Cabinet Office (1-6-1 Nagata-cho, Chiyoda-ku, Tokyo 100-8914, Japan. TEL: +81- (0)3-6257-1628).

**Funding:** The authors received no specific funding for this work.

**Competing interests:** The authors have declared that no competing interests exist.

daylight. This SAD hypothesis, if true, is important both in economics and psychology because it is a typical example challenging the mainstream asset pricing models building upon strict economic rationality [15] and is one of the few indications of the effect of emotions on financial decision making in the real world [16–18]. To elucidate the importance of this topic, Kamstra, Kramer & Levi [2], the seminal study on this topic, is cited in 1087 papers according to Google scholar as of November 20, 2020.

However, recent studies in psychology cast doubt on the existence of SAD [19–21]. In particular, a recent systematic review failed to find any consistent evidence of a seasonal depressive cycle at the population level, mainly due to the heterogeneity among extracted studies [22]. Thus, attributing the cause of seasonal asset price transitions to the seasonal emotional cycle is losing foundation and many studies relying on the SAD hypothesis are in jeopardy. Although the existence of the seasonal emotional cycle should be explored with rigorous methodology [22], there seems to be little hope for this in the near future. Hence, alternative ways to deal with this issue are urgently required. An option is to look for evidence of an optimism/ pessimism cycle or risk seeking/avoidance cycle based on the changes in the length of daylight, preferably using individual data. Consumer confidence [23], people's outlook for future economy, can be a mediator linking the length of the daylight to the seasonal asset price transitions. Thus, the purpose of the present study is to examine if there is a positive association between consumer confidence and length of daylight.

## A brief review of the relevant literature and controversies

The claim that SAD causes seasonal fluctuations in asset prices was spearheaded by Kamstra et al. [2] on stock prices. They claimed that: i) depressed people tend to be more pessimistic and risk-averse; ii) from autumn to winter, many people experience depressive symptoms (SAD), or their milder versions (winter blues), owing to the diminishing length of daylight; this causes people to become risk averse; and iii) as a result of this risk aversion, people tend to sell stocks in autumn that are considered risky assets, causing stock prices to fall. After the winter solstice, with increasing length of daylight, people's risk aversion diminishes and they begin to buy stocks, causing a rise in stock prices. According to Kamstra et al. [2], this trend is more salient in areas at a high latitude, where there are large differences between the length of daylight during summer and winter. Inspired by the research conducted by Kamstra et al. [2], seasonal variations consistent with SAD have been reported in the pricing of initial public offerings [3, 5, 6], art auction prices [7], real estate prices [4], bond prices [8], as well as stock prices in several countries [9, 10, 13]. However, other studies have remained dubious regarding the effect of SAD on stock prices [24–27]. A major criticism is that the studies claiming the effect of SAD on asset price transitions just confirm the association between the length of daylight and asset prices and the involvement of SAD has not been directly examined, leaving the causal mechanisms linking length of daylight to asset prices unexplored [18, 25]. In addition to these criticisms, the existence of SAD itself has been challenged by recent studies in psychology [14, 22]. Several studies denied the existence of a seasonal depressive cycle [19–21], whereas others supported the existence of SAD [28–30]. A recent study suggested that only highly neurotic people experience a seasonal emotional transition, and the seasonal emotional transition at the population level is small even if it exists [31].

## Overall aim of the present study

Against this background, the purpose of the present study is to examine if there really is any possible psychological mediator linking length of daylight to seasonal asset price transitions. One major missing part of previous studies on the relationship between length of daylight and

asset price transition is the exploration of the relationship between the optimism/pessimism cycle and the length of daylight. Optimism/pessimism is linked to risk seeking/avoidance [32–34]. Thus, if there is an association between optimism/pessimism and the length of daylight, it can be a possible cause of seasonal asset price transitions [11]. The exploration of the relationship between the optimism/pessimism cycle and length of daylight can also provide indirect evidence for the existence of the seasonal cycle of depression since pessimism is linked to depression [35–37].

In the present study, we shed light on the Japanese version of the consumer confidence index (CCI), an indicator of households' optimism/pessimism for future economy. The CCI is derived from the Consumer Confidence Survey (CCS) by the Japanese government. It is a monthly survey of Japanese households. The individual longitudinal data (panel data) exists for each month, as the same households consecutively respond to the survey for 12 to 15 months. The CCI was evaluated using four questions that assessed the respondents' subjective outlook regarding the economy six months into the future. The CCS also assessed one's expectations regarding the future value (six months later) of one's assets.

If there is seasonal fluctuation—high in summer and low in winter—reflecting the changing length of daylight in CCI or in asset value expectations (AVE), and it is more striking in high-latitude areas, it provides evidence for the existence of a seasonal cycle of optimism and pessimism for future economy. This seasonal cycle, if it exists, can be a possible cause of seasonal asset price transitions and offers indirect evidence for the existence of the seasonal cycle of depression. In the present study, we hypothesized that CCI and AVE are positively associated with the length of daylight.

## Materials and methods

### CCI and AVE

The CCS is carried out by the Cabinet Office of the Japanese government every month nationwide. CCI and AVE are calculated based on answers from the CCS. For the present study, we obtained raw CCS data from the Cabinet Office for research purposes. The CCS began in 1957 and was designed to quickly understand shifts in consumer perceptions. Originally, the survey was conducted only in March, June, September, and December, but since April 2004, the survey has been conducted every month. Hence, we only used data after April 2004 until August 2018. In addition to the questions on the CCI and AVE explained later, the CCS was composed of questions on the prediction of daily purchasing goods' prices one year ahead, the sex of the household heads, whether they are in labor force, the types of jobs they take (agriculture, being employed, self-employed, or others), the age of the household heads, the number of household members, the number of those who are in labor force within the household, the number of unemployed and job-seeking persons within the household, the annual household income (seven categories), the main income categories (i.e., salary, business, pension, or others), the types of residence (i.e., self-owned independent housing, self-owned condominium unit, public rental housings, residence for employees, and private rental housings), whether they have housing mortgage, and the floor space of the residences.

**Survey participants.** A three-level stratified sampling was used for the sampling procedure in the CCS. Before March 2013, the sample size included 6,720 households. Since April 2013, 8,400 households participated. The sampling procedure since April 2013 has been made public (https://www.esri.cao.go.jp/jp/stat/shouhi/shouhi_gaiyou.html#a2). At the first stage, 242 cities, towns, or villages from all 47 prefectures were selected, according to the results of the national census carried out every five years (the latest one occurred on Oct. 1st, 2015). This process was mostly random except that two cities were selected in scarcely populated

prefectures, in which the calculated number was one. At the second stage, 336 local units were randomly selected. At the third stage, 25 households were randomly selected in each of the local units. Before May 2006, participating households were classified into four groups and they answered the questionnaire consecutively for 12 months; they were then replaced by a newly created group. Since June 2006, participants were classified into five groups and they answered the questionnaire consecutively for 15 months; they also were replaced by a newly created group. The CCS was implemented on the 15th day of each month. There were 964,361 total responses for 85,753 households and each household responded, on average, 11.2 times (average response rate: 75.8%). S1 Table shows the basic statistics of households who answered the survey by prefecture, spanning April 2004 to August 2018. As it is not obligatory that household heads themselves answer the CCS, the sex of those who answered the CCS cannot be identified.

**CCI.**   CCI was calculated based on four questions in the CCS. These questions assessed expectations of overall livelihood, income growth, employment, and willingness to buy durable goods over the next six months. Each answer was rated on a scale of 1 (improve) to 5 (worsen). Answers of 1 (improve), 2 (improve slightly), 3 (no change), 4 (worsen slightly), and 5 (worsen) were converted to 100, 75, 50, 25, and 0, respectively. To calculate CCI as a whole, the converted scores were averaged. For example, if a respondent chose 3 for all four questions, the respondent's CCI would be 50. Higher scores on CCI indicate a higher level of consumer confidence.

**AVE.**   In the CCS, the AVE question is separate from CCI questions. It asks: "Do you think that the value of assets your household owns, such as stock and land, will increase or decrease in the coming half a year?" Its answer was rated on a scale from 1 (increase) to 5 (decrease). Answers of 1 (increase), 2 (increase slightly), 3 (no change), 4 (decrease slightly), and 5 (decrease) were converted to 100, 75, 50, 25, and 0, respectively. As this question is closely related to price expectation on stocks and real estates, we used AVE as an outcome variable of interest, as well as CCI. The means of CCI and AVE from April 2004 to August 2018 were 42.02 and 42.24, respectively (S1 Table). Fig 1 graphically shows the transitions of CCI and AVE for each month.

### Length of daylight

Length of daylight for each day was calculated using the formula by Kamstra et al. [2], which is based on the latitude of the region and days since January 1st. For latitude, we used the value of the city hall in which each household resided. As a measure of SAD, Kamstra et al. [2] used hours of nighttime from autumn equinox to spring equinox as their independent variable of interest. However, we used length of daylight for the entire year, as we took into account that recent psychological studies have shown that seasonal emotional cycles are observed, not only in autumn and winter, but also in spring and summer, and that the lowest level of depression is observed in summer or early autumn [19, 29]. This practice was also carried out by Kliger et al. [7]; hence, we used length of daylight from dawn to sunset as one of our independent variables.

### Climate data

For climate factors, we used cloud cover, precipitation, and temperature. We collected this information for the prefectural capitals from the Japan Meteorological Agency's website (https://www.data.jma.go.jp/gmd/risk/obsdl/index.php). We used an average of five days between the 11th and 15th for cloud cover, precipitation, and temperature data, considering that the CCS is implemented on the 15th day of each month. Cloud cover ranged from zero to ten, with higher values indicating more cloud coverage. Precipitation was measured via

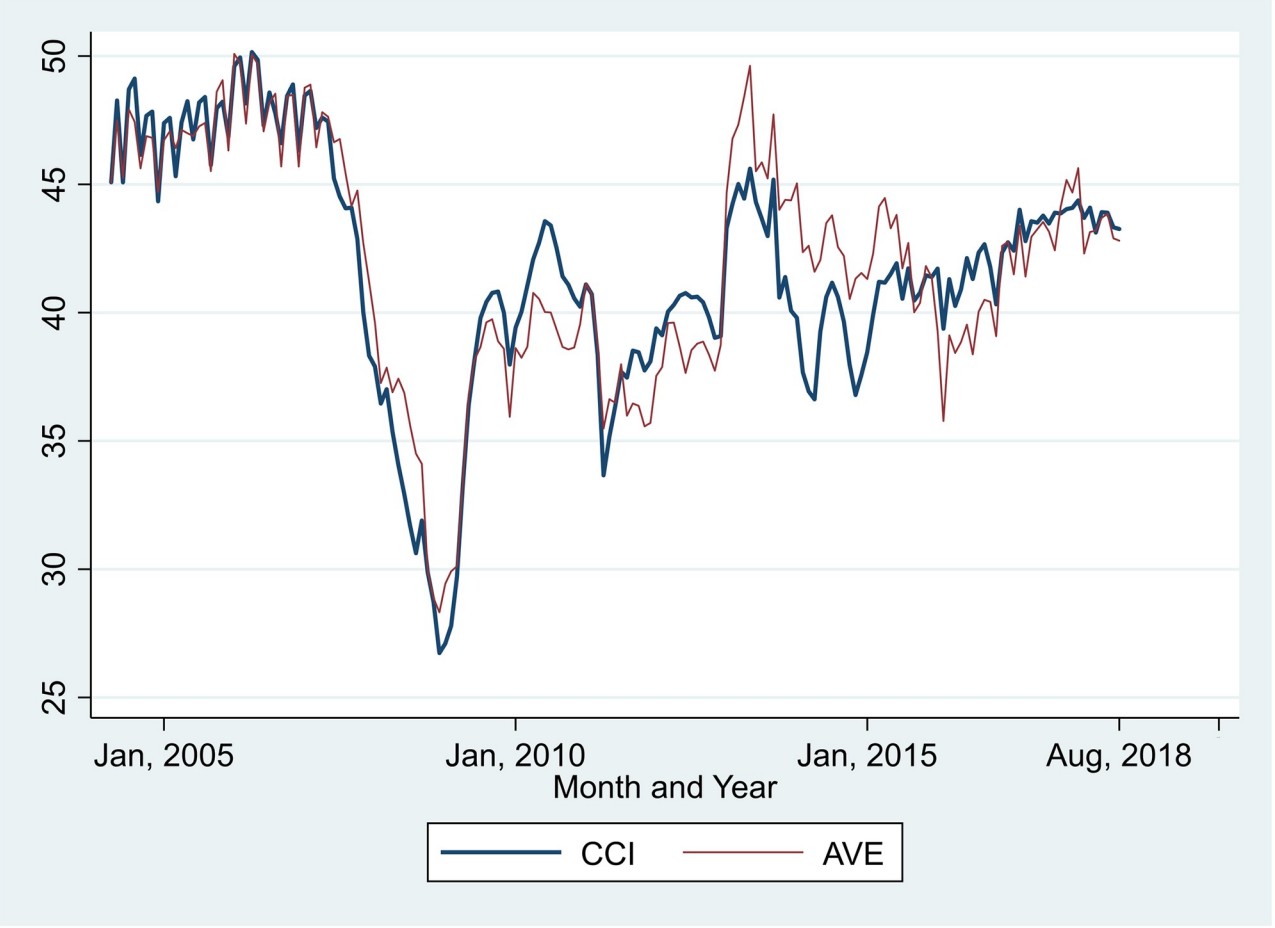

**Fig 1. CCI and AVE transitions.**

millimeters per day, and temperature was based on the Celsius temperature scale. For Saitama, Chiba, Shiga, and Yamaguchi Prefecture, cloud cover data from their prefecture capitals were missing; hence, for all the three measures, we used data from Kumagaya, Choushi, Hikone, and Shimonoseki City, which belong to the same prefectures, respectively.

## Analysis strategy

We estimated the associations between length of daylight and CCI using four regression models. Models 1 and 2 were intended to examine whether there is a seasonal calendar cycle of CCI compatible with changing length of daylight. Model 3 was intended to directly examine whether length of daylight is positively associated with CCI. Model 4 was intended to examine whether regions with higher latitude experience a more striking seasonal cycle of CCI. AVE was estimated in the same way as CCI.

For Model 1, we estimated monthly CCI transitions as follows:

$$CCI_{it} = \alpha_i + \beta_{JANUARY}JANUARY_t + \beta_{FEBRUARY}FEBRUARY_t + \beta_{MARCH}MARCH_t + \beta_{APRIL}APRIL_t \\ + \beta_{MAY}MAY_t + \beta_{JUNE}JUNE_t + \beta_{JULY}JULY_t + \beta_{AUGUST}AUGUST_t \\ + \beta_{SEPTEMBER}SEPTEMBER_t + \beta_{OCTOBER}OCTOBER_t + \beta_{NOVEMBER}NOVEMBER_t + u_{it} \quad (1)$$

The variables were defined as follows. $CCI_{it}$ represents CCI of household $i$ at time $t$ ($t$ denotes monthly level data, spanning April 2004 to August 2018). $\alpha_i$ is an unobservable household-specific characteristic that differs from household to household, but does not vary over time. $u_{ti}$ is the error term. The definitions of these three variables are the same in the other models. $JANUARY_t$ is a binary variable in which the value is 1 if $t$ belongs to January and 0 if otherwise. This treatment is the same for $February_t$ through $November_t$. The reference category is December when length of daylight is the shortest of all months. If length of daylight is positively associated with CCI, $\beta_{JANUARY}$ through $\beta_{NOVEMBER}$ should all be positive and $\beta_{JUNE}$ is expected to be the largest among them.

For Model 2, we constructed the CCI seasonal trends applying a cosinor model. This model is also used in Lyall et al. [29] to examine the SAD cycle. We followed the same methodology shown in Lyall et al. [29] and Bernett and Dobson [38], as follows:

$$CCI_{it} = \alpha_i + \beta_{COS}\cos(\omega_t) + \beta_{SIN}\sin(\omega_t) + u_{it} \tag{2}$$

where $\omega_t$ is calculated as follows:

$$\omega_t = \frac{2\pi(day_t - 1)}{365 \ or \ 366} \tag{3}$$

where $day_t$ is the days of the year since January 1st for time $t$, which are 15, 46, 75, 106, 136, 167, 197, 228, 259, 289, 320, and 350 for each month for non-leap years. One is added to the above numbers in leap years after March. The denominator for $\omega_t$ is 366 in leap years and 365 in other years. Amplitude (A; half of the height of the cycle) was calculated as follows:

$$A = \sqrt{\hat{\beta}_{COS}^2 + \hat{\beta}_{SIN}^2} \tag{4}$$

Acrophase ($\varphi$; days from January 1st to the first peak of the cycle) was calculated as follows:

$$\varphi = 365 * \frac{tan^{-1}(\hat{\beta}_{SIN}/\hat{\beta}_{COS})}{2\pi} + 1 \tag{5}$$

Standard errors (SEs) and confidence intervals (CIs) for amplitude and acrophase were calculated using the delta method. If length of daylight is positively associated with CCI, amplitude is expected to be significantly positive and acrophase should be close to the summer solstice in summer.

For Model 3, we observed the association between CCI and length of daylight as follows:

$$CCI_{it} = \alpha_i + \rho_1 CCI_{it-1} + \rho_2 CCI_{it-2} + \beta_{DAYLIGHT}DAYLIGHT_{jt} + \beta_{CLOUD}CLOUD_{kt}$$
$$+ \beta_{PRECIPITATION}PRECIPITATION_{kt} + \beta_{TEMPERATURE}TEMPERATURE_{kt} + u_{it} \tag{6}$$

The variables were defined as follows; $CCI_{it-1}$ and $CCI_{it-2}$ are lagged CCI variables, one month before and two months before, respectively. $DAYLIGHT_{jt}$ is length of daylight at the latitude of the city hall (j) in the residential area of each survey household at the time of each survey. $CLOUD_{kt}$, $PRECIPTION_{kt}$, and $TEMPERATURE_{kt}$ are the cloud cover, precipitation, and temperature of the prefectural office in the prefecture (k) where each survey household resided at the time of the survey. One problem with this model is that length of daylight is related to cloud cover, precipitation, and temperature. In particular, the temperature is directly affected by length of daylight, but the reverse is not possible. If the variables of cloud cover, precipitation, and temperature are put into the estimation formula, there is a possibility of over-adjustment. Therefore, we also estimated the association between CCI and daylight length without

adjusting for cloud cover, precipitation, and temperature. If length of daylight is positively associated with CCI, $\beta_{DAYLIGHT}$ is expected to be significantly positive.

For Model 4, to estimate the association between latitude and CCI, CCI was regressed on latitude, season (days from the winter solstice), and their interaction, as follows.

$$CCI_{it} = a_i + \beta_{LATITUDE}LATITUDE_j + \beta_{SEASON}SEASON_t + \beta_{LATITUDE*SEASON}LATITUDE_j * SEASON_t + u_{it} \quad (7)$$

Variables were defined as follows. $LATITUDE_j$ is the latitude of each city hall for each household's residential city. $SEASON_t$ is the number of days from the winter solstice to the survey day of each month. $LATITUDE_j * SEASON_t$ is the interaction term of $LATITUDE_j$ and $SEASON_t$. As the survey day is the 15th day of each month, $SEASON_t$ is 24, 55, 83, 114, 144, 175, 159, 129, 97, 67, 36, and 6 for each month of non-leap years, beginning in January. We constructed $SEASON_t$ following Kerr et al. [19] and Traffanstedt et al. [20]. If length of daylight is positively associated with CCI, $\beta_{LATITUDE * SEASON}$ is expected to be significantly positive.

For all models, we estimated these equations using fixed-effects models with panel data to control for $\alpha_i$, or the time-invariant unobserved heterogeneity across households [39]. Significance level was set at $p < 0.05$. We used STATA 15 to conduct our analyses.

## Results

### Descriptive statistics

Table 1 shows the monthly data for CCI and AVE from April 2004 until August 2018, with the highest values occurring in May.

### Monthly estimates of CCI and AVE (Model 1)

Estimates of Model 1 are shown in Table 2. The graphic monthly estimates of CCI and AVE for each year, based on the fixed-effects model, are shown in Fig 2. The estimates of CCI and AVE from April 2004 to August 2018 were the lowest in December when the length of daylight was the shortest. However, the highest CCI and AVE estimates were observed in May, instead of June when length of daylight was the longest. After peaking in May, CCI and AVE both declined in June and rose again in July. Afterward, both CCI and AVE continued to decline

**Table 1. Monthly statistics of CCI and AVE from April 2004 to August 2018.**

|  | CCI | | | AVE | | |
|---|---|---|---|---|---|---|
|  | n | Mean | SD | n | Mean | SD |
| January | 77,211 | 41.84 | (15.10) | 77,242 | 42.54 | (17.98) |
| February | 77,404 | 41.61 | (15.13) | 77,430 | 41.80 | (18.03) |
| March | 76,931 | 41.65 | (15.21) | 76,961 | 42.05 | (19.19) |
| April | 84,927 | 41.94 | (15.05) | 85,007 | 42.65 | (18.03) |
| May | 84,953 | 42.92 | (14.58) | 85,006 | 43.23 | (17.33) |
| June | 84,152 | 42.43 | (14.72) | 84,205 | 42.58 | (18.62) |
| July | 84,243 | 42.79 | (14.45) | 84,265 | 42.80 | (17.14) |
| August | 83,955 | 42.70 | (14.36) | 83,978 | 42.56 | (17.09) |
| September | 77,619 | 42.21 | (14.62) | 77,650 | 41.83 | (18.54) |
| October | 77,808 | 41.88 | (14.57) | 77,841 | 41.82 | (17.60) |
| November | 77,601 | 41.45 | (14.91) | 77,624 | 41.71 | (17.89) |
| December | 77,098 | 40.55 | (15.46) | 77,113 | 41.09 | (19.29) |

SD = Standard Deviation, CCI = Consumer Confidence Index, AVE = Asset Value Expectation.

**Table 2. Fixed-effects model estimation of monthly CCI and AVE from Model 1.**

| | CCI | AVE |
|---|---|---|
| $\beta_{JANUARY}$ | 1.186*** (0.043) | 1.359*** (0.068) |
| $\beta_{FEBRUARY}$ | 0.871*** (0.046) | 0.542*** (0.071) |
| $\beta_{MARCH}$ | 1.121*** (0.049) | 0.981*** (0.073) |
| $\beta_{APRIL}$ | 0.987*** (0.050) | 1.196*** (0.073) |
| $\beta_{MAY}$ | 1.964*** (0.050) | 1.787*** (0.072) |
| $\beta_{JUNE}$ | 1.666*** (0.050) | 1.288*** (0.073) |
| $\beta_{JULY}$ | 1.956*** (0.050) | 1.464*** (0.072) |
| $\beta_{AUGUST}$ | 1.871*** (0.049) | 1.256*** (0.071) |
| $\beta_{SEPTEMBER}$ | 1.649*** (0.048) | 0.700*** (0.072) |
| $\beta_{OCTOBER}$ | 1.273*** (0.046) | 0.667*** (0.070) |
| $\beta_{NOVEMBER}$ | 0.789*** (0.043) | 0.521*** (0.068) |
| Intercept | 40.726*** (0.030) | 41.247*** (0.051) |
| No. of observations | 963,902 | 964,322 |
| No. of groups | 85,740 | 85,753 |
| R-squared (within) | 0.004 | 0.001 |
| R-squared (between) | 0.002 | 0.003 |
| R-squared (overall) | 0.002 | 0.001 |

CCI = Consumer Confidence Index, AVE = Asset Value Expectation.

*** $p < 0.1\%$. Robust standard errors are in parentheses. CCI and AVE were indexed based on the formula from the Cabinet Office of Japan.

until December. Since the questions used to assess CCI and AVE did not ask for the perception of economic conditions at that time, but did ask about one's outlook after six months, based on Table 2 and Fig 2, households expected that the economy and asset values would deteriorate from January (six months after July) to June (six months after December) on average. Participants were most economically optimistic in May, became pessimistic from August to December, and began to be optimistic again in January. Except for January and June, the results are compatible with the hypothesis that length of daylight is positively associated with CCI and AVE.

## Seasonal cycle of CCI and AVE as Estimated by the cosinor model (Model 2)

Estimates of Model 2 are shown in Table 3 and a graphical prediction of CCI and AVE from the cosinor model is shown in Fig 3. Results indicated that amplitude was significantly positive for both CCI and AVE. Acrophases for CCI and AVE were 187.8755 days from January 1st (July 7th, 95%CI: July 4th–July 10th) and 153.4521 days from January 1st (June 3rd, 95%CI: May 28th–June 7th), respectively. Although the 95% CIs do not include the summer solstice, they are close to it; hence, estimates using the cosinor model show that the seasonal cycle was more or less compatible with the hypothesis that length of daylight is positively associated with CCI and AVE, but not perfectly.

## Length of daylight and CCI and AVE (Model 3)

Estimates of Model 3 are shown in Table 4. Length of daylight was positively correlated with CCI and AVE. However, $\beta_{TEMPERAURE}$ was not significant for CCI but was negatively significant for AVE, suggesting that CCI and AVE were positively associated with length of daylight, but not with temperature.

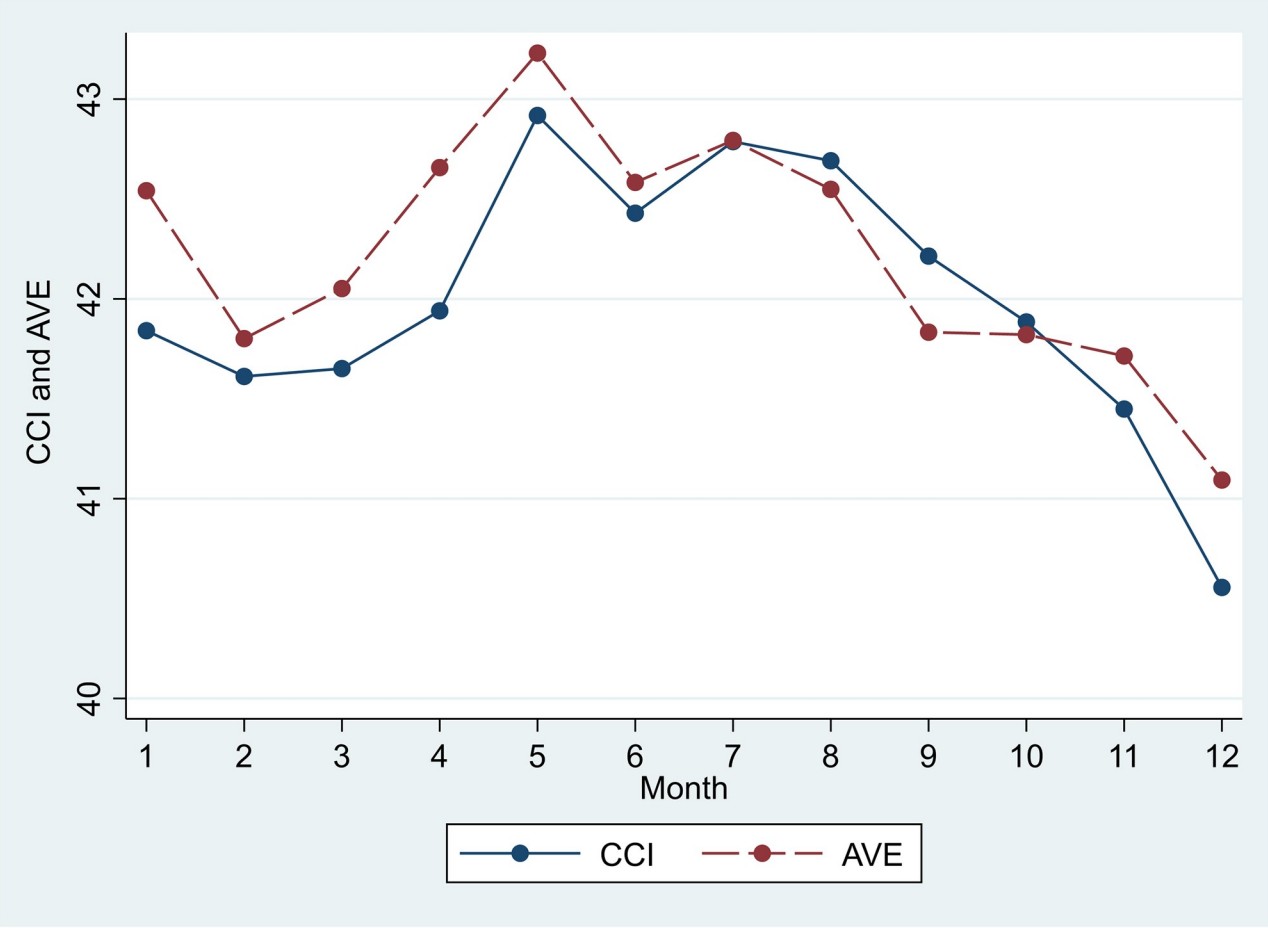

**Fig 2. Estimates of the monthly CCI and AVE from Model 1.**

**Table 3. Fixed-effects model estimation of CCI and AVE through cosinor model (Model 2) with amplitude and acrophase.**

|  | CCI | AVE |
|---|---|---|
| $\beta_{cos}$ | -0.665*** (0.019) | -0.462*** (0.023) |
| $\beta_{SIN}$ | -0.050*** (0.018) | 0.263*** (0.023) |
| Intercept | 42.003*** (0.000) | 42.23*** (0.001) |
| No. of observations | 963,902 | 964,322 |
| No. of groups | 85,740 | 85,753 |
| R-squared (within) | 0.003 | 0.001 |
| R-squared (between) | 0.003 | 0.003 |
| R-squared (overall) | 0.001 | 0.001 |
| Amplitude | 0.666*** (0.186) | 0.531*** (0.234) |
| Acrophase | 187.876*** (1.586) | 153.4521*** (2.546) |

CCI = Consumer Confidence Index, AVE = Asset Value Expectation.

*** $p < 0.1\%$. Robust standard errors are in parentheses except for amplitude and acrophase. For these, standard errors were calculated using the delta method. CCI and AVE were indexed based on the formula from the Cabinet Office of Japan.

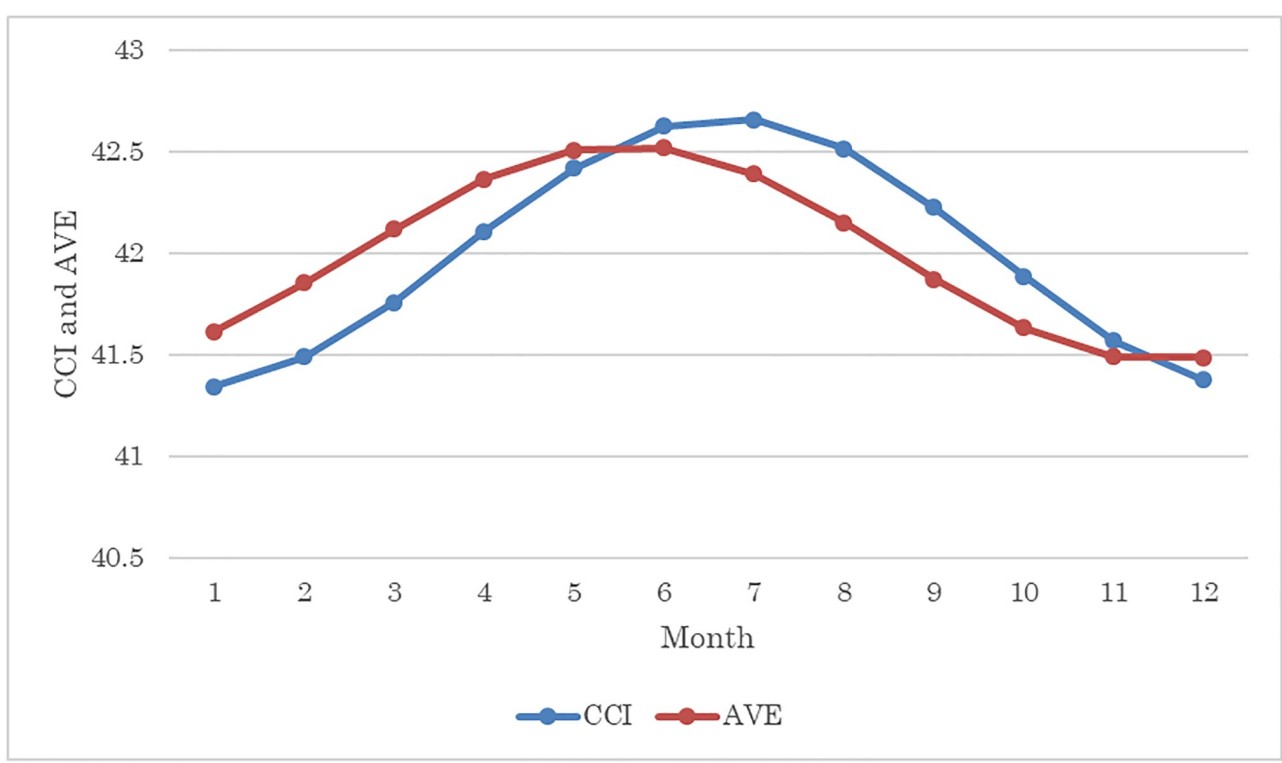

**Fig 3. Predicting CCI and AVE through the cosinor model.**

### Latitude, CCI, and AVE (Model 4)

Estimates of Model 4 are shown in Table 5. Of note, $\beta_{LATUTUDE}$ was omitted from Model 4 because we applied a fixed-effects model, whereby time-invariant variables (in this case, $LATITUDE_j$) were omitted. Results indicated that $\beta_{LATUTUDE\,*\,SEASON}$ was significantly positive; thus, this model supported the hypothesis that length of daylight was positively associated with CCI and AVE

**Table 4. Fixed-effects model estimation of CCI and AVE with length of daylight, cloud cover, precipitation, and temperature (Model 3).**

|  | CCI | CCI | AVE | AVE |
|---|---|---|---|---|
| $\rho_1$ | 0.203*** (0.002) | 0.203*** (0.002) | 0.055*** (0.002) | 0.055*** (0.002) |
| $\rho_2$ | 0.060*** (0.002) | 0.060*** (0.002) | -0.002 (0.002) | -0.002 (0.002) |
| $\beta_{DAYLIGHT}$ (per hour) | 0.226*** (0.007) | 0.194*** (0.009) | 0.191*** (0.010) | 0.275*** (0.014) |
| $\beta_{CLOUD}$ (per one point) |  | 0.070*** (0.006) |  | 0.020* (0.009) |
| $\beta_{PRECITITATION}$ (per 1mm/day) |  | -0.001*** (0.000) |  | 0.000 (0.001) |
| $\beta_{TEMPERATURE}$ (per ˚C) |  | 0.004 (0.002) |  | -0.026*** (0.003) |
| Intercept | 28.194*** (0.132) | 28.086*** (0.144) | 37.636*** (0.171) | 36.883*** (0.192) |
| No. of groups | 75,645 | 75,645 | 75,660 | 75,660 |
| R-squared (within) | 0.055 | 0.055 | 0.004 | 0.004 |
| R-squared (between) | 0.933 | 0.933 | 0.850 | 0.843 |
| R-squared (Overall) | 0.575 | 0.575 | 0.260 | 0.255 |

CCI = Consumer Confidence Index, AVE = Asset Value Expectation.

* $p < 5\%$.

*** $p < 0.1\%$. Robust standard errors are in parentheses. CCI and AVE were indexed based on the formula from the Cabinet Office of Japan.

**Table 5. Fixed-effects model to predict CCI and AVE from season and latitude (Model 4).**

|  | CCI | AVE |
|---|---|---|
| $\beta_{SEASON}$ | -0.0041 (0.0038) | -0.0090* (0.0045) |
| $\beta_{LATUTUDE}$ | Omitted | Omitted |
| $\beta_{LATUTUDE * SEASON}$ | 0.0004*** (0.0001) | 0.0004*** (0.0001) |
| Intercept | 41.1980*** (0.0231) | 41.6167*** (0.0293) |
| No. of observations | 963,902 | 964,322 |
| No. of groups | 85,740 | 85,753 |
| R-squared (within) | 0.0025 | 0.0007 |
| R-squared (between) | 0.0015 | 0.0006 |
| R-squared (overall) | 0.0011 | 0.0003 |

CCI = Consumer Confidence Index, AVE = Asset Value Expectation.

* $p < 5\%$.

*** $p < 0.1\%$. Robust standard errors are in parentheses. CCI and AVE were indexed based on the formula from the Cabinet Office of Japan.

## Robustness checks

We performed two robustness checks. First, we divided the whole periods into two. The data we used in the present study is from April 2004 to August 2018, for a total of 173 months. The first period was from April 2004 to March 2011 (84 months) and the second was from April 2011 to August 2018 (89 months). The monthly summary statistics of CCI and AVE for the two periods are shown in S2 Table. The results of Model 1 are shown in S3 Table and graphically shown in S1 Fig. The results of Model 2 are shown in S4 Table and graphically shown in S2 Fig. The results of Model 3 are shown in S5 Table for CCI and S6 Table for AVE. The results of Model 4 are shown in S7 Table. In general, the seasonal pattern is less clear and smaller in the latter period, as shown in S1 and S2 Figs. However, most key coefficients (such as $\beta_{DAYLIGHT}$ in Model 3 (S5 Table) and $\beta_{LATUTUDE * SEASON}$ in Model 4 (S7 Table)) remained significant except $\beta_{LATUTUDE * SEASON}$ in AVE until March 2011 (S7 Table).

In the second robustness check, we divided the samples into two according to the latitude by the median. The total number of participants was 85,730. The latitude of the first group was below 35.41 and the latitude of the second group was above 35.41. In the lower latitude areas, there were 43,289 participants and the mean latitude for them was 34.133. In the higher latitude areas, there were 42,464 participants and the mean latitude for them was 37.060. The monthly summary statistics of CCI and AVE for the two groups are shown in S8 Table. The results of Model 1 are shown in S9 Table and graphically shown in S3 Fig. The results of Model 2 are shown in S10 Table and graphically shown in S4 Fig. The results of Model 3 are shown in S11 Table for CCI and S12 Table for AVE. The results of Model 4 are shown in S13 Table. In general, the seasonal pattern remains stable regardless of the latitude (S3 and S4 Figs). $\beta_{DAYLIGHT}$ in Model 3 (S11 and S12 Tables) were significant. $\beta_{LATUTUDE * SEASON}$ in Model 4 for both CCI and AVE in higher latitude areas were not significant (S13 Table).

## Discussion

The present study examined whether Japan's CCI and AVE are positively associated with length of daylight. Analyzing the individual longitudinal data of the CCI and AVE, which was constructed from raw data obtained from 2004 to 2018, our results indicated that, (i) there was

a seasonal cycle of CCI and AVE; the trough occurred in December and the peak in early summer; (ii) the length of daylight was positively associated with CCI and AVE; and (iii) the higher the latitude, the larger the seasonal cycle of CCI and AVE. These results support the hypothesis that Japan's CCI and AVE are positively associated with the length of daylight.

As far as we know, this is the first study to examine the association between consumer confidence and the length of daylight using individual longitudinal data to construct consumer confidence. The causal direction from CCI and AVE to the length of daylight is implausible because the latter cannot be artificially controlled. Also, it is hard to find any confounder that would affect both the length of daylight and CCI (AVE). Hence, our results suggest that there is a causal direction from the length of daylight to CCI and AVE. However, finding any mediator from the length of daylight to CCI (AVE) is difficult and beyond the scope of the present study.

Although the CCI and AVE are somehow associated with the length of daylight, January and June are exceptions. The hike of the CCI and AVE in January, accompanied by a fall in February, may be the results of the effect of the New Year celebrations, which is reflective in stock prices [11, 40, 41]. Another anomaly is observed in June, when a sudden drop of CCI (AVE) was followed by a sudden rise in July. Although the reason is unclear, Japan's annual rainy season (tsuyu), which occurs mainly in June, may be responsible for this anomaly. This time of the year is rather humid and it is uncomfortable for many people in Japan. This calendar factor may have modified the peak of the seasonal variation, although this hypothesis needs to be verified in further research.

Our empirical results suggest that the seasonal cycle of optimism/pessimism for future economy is one of the causes of the seasonal cycle of asset prices, including stock prices. Several studies showed that consumer confidence is reflected in stock prices [42–44]. For example, Lemmon and Portniaguina [42] showed that the returns of stocks with low institutional ownership, as well as small stocks, are forecast by consumer confidence. More specifically, Rojo-Suárez and Alonso-Conde [44] showed that consumer confidence in Japan affects the stock prices of the Tokyo Stock Exchange. Based on these studies, it seems reasonable to assume that the seasonal cycle of people's optimism/pessimism exemplified by the CCI (AVE) is one of the causes of asset price transitions in Japan. Supporting this assumption, Sakakibara et al. [9] showed that stock prices go up from January to June and go down from July to December, which is compatible with the findings of the present study.

However, extending the results of the present study to countries other than Japan need caution. The "sell in May" effect suggests that stock prices go down before the summer solstice and the Halloween effects suggests that stock prices begin to go up before the winter solstice [1, 12]. These effects cannot be solely explained by the length of daylight. It is not clear whether the effects of the length of daylight on stock prices are totally denied or both the length of daylight and other factors are simultaneously involved in many countries [11, 12]. The existence of seasonal optimism/pessimism cycles in countries other than Japan and their relationships with asset price transitions, and with stock prices in particular, needs to be further explored.

Although our results suggest that there is a causal direction from the length of daylight to CCI and AVE, the mechanisms between them are still unclear. One possible mechanism is the seasonal emotional cycle exemplified by SAD. Causal directions from emotions to perceptions have been already shown, in particular from depression to pessimism [16, 45, 46]. Although several explanations have been proposed for seasonal emotional changes, including physical activity [47], melatonin secretion [48], and vitamin D level [49], the existence of the seasonal emotional cycle is still debated [22]. Given this uncertainty and the already reported correlation between pessimism and depression [45], one of the contributions of the present study is that the existence of an emotional seasonal cycle compatible with the length of daylight may be assumed from the seasonal optimism/pessimism cycle associated with the length of daylight

shown in the present study. A recent study reported a positive association between consumer sentiment and mental stress [50]. Similarly, several studies confirmed the associations between the expectation of future economy and happiness/mental health [51–53]. These studies along with the present study suggest the existence of an emotional seasonal cycle in line with the existence of an optimism/pessimism cycle.

The results from the robustness checks should be interpreted cautiously. When the data were split by time, the seasonal patterns were less clear and smaller in the latter period compared to the former one. One explanation is that CCI and AVE are moved by various events in the short term, whereas the effects of the daylight length is rather stable and easier to observe in the long-term data, in which the effects of various short-term events are cancelled out. By contrast, dividing the data by the latitude showed similar patterns for both lower and higher latitude areas. If CCI and AVE were affected by the length of daylight, differences of CCI and AVE between winter and summer should be larger in the higher latitude areas than the lower latitude areas. Indeed, the amplitude of CCI and AVE were larger in the higher latitude area group than the lower latitude area group, but the difference was not significant (S10 Table and S4 Fig). One possible explanation is that the difference in the latitude between the two groups is too small to show any significant effect. The mean latitude was 34.133 for the lower latitude group and 37.060 for the higher latitude group. This difference of around 300 km may be too small to find the difference. Studies in other countries are required to further explore this.

There are several strengths in the present study. First, this study used raw data of the CCS, which are longitudinal data (panel data), thus making it possible to control time-invariant heterogeneity unlike in cross-sectional studies. Second, we examined the relationship between the length of daylight and CCI and AVE through four models, which contributes to ensure the robustness of our findings.

There are some limitations in the present study. First, the adjusted R-squared for the models, except Model 3, are below 0.01, suggesting that the explanatory power is quite low. It is high for Model 3, but this reflects the fact that the CCIs of the previous months are included as explanatory variables in the model.

Secondly, the present study is limited to Japan only; thus, our study findings should be generalized with caution. Similar studies in other countries are necessary to understand this relationship in greater depth. If the length of the daylight does really affect the consumers' confidence levels, similar results should be obtained in temperate countries other than Japan. In particular, if fluctuations of consumer confidence, which is at the highest level in December and lowest in June, are found in the southern hemisphere, our findings will be supported in a more general context. Third, given the lack of individual longitudinal data recording both asset purchasing behavior and consumer confidence, we could not provide direct evidence of whether seasonal changes of consumer confidence lead to changes in the asset purchasing behavior at the micro data level. In future studies, the construction of individual longitudinal data merging asset purchasing behavior data and consumer confidence data, as well as data on emotions, including depression, are highly recommended.

## Conclusions

Analyzing the individual longitudinal data of CCI and AVE generated from raw data between 2004 and 2018, we found a seasonal cycle. The trough of this cycle occurred in December and the peak in the early summer. We also found that the higher the latitude, the larger the seasonal cycle of CCI and AVE. These results suggest that differences in the length of daylight may cause asset price transitions through the seasonal optimism/pessimism cycle as shown by the

CCI and AVE. To further explore the causal relationship between these variables and draw reliable policy implications, the construction of individual longitudinal data merging asset purchasing behavior data and consumer confidence data is needed.

## Supporting information

**S1 Fig. Estimates of the monthly CCI and AVE from Model 1 for the two periods.**
(TIF)

**S2 Fig. Predicting CCI and AVE through the cosinor model (Model 2) for the two periods.**
(TIF)

**S3 Fig. Estimates of the monthly CCI and AVE from Model 1 for the lower and higher latitude areas.**
(TIF)

**S4 Fig. Predicting CCI and AVE through the cosinor model (Model 2) for the lower and higher latitude areas.**
(TIF)

**S1 Table. Summary Statistics of Participating Households from the Consumer Confidence Survey (CCS).**
(DOCX)

**S2 Table. The monthly summary statistics of CCI and AVE for the two periods.**
(DOCX)

**S3 Table. Fixed-effects model estimation of monthly CCI and AVE from Model 1 for the two periods.**
(DOCX)

**S4 Table. Fixed-effects model estimation of CCI and AVE through cosinor model (Model 2) with amplitude and acrophase for the two periods.**
(DOCX)

**S5 Table. Fixed-effects model estimation of CCI with length of daylight, cloud cover, precipitation, and temperature (Model 3) for the two periods.**
(DOCX)

**S6 Table. Fixed-effects model estimation of AVE with length of daylight, cloud cover, precipitation, and temperature (Model 3) for the two periods.**
(DOCX)

**S7 Table. Fixed-effects model to predict CCI and AVE from season and latitude (Model 4) for the two periods.**
(DOCX)

**S8 Table. The monthly summary statistics of CCI and AVE for the lower and higher latitude areas.**
(DOCX)

**S9 Table. Fixed-effects model estimation of monthly CCI and AVE from Model 1 for the lower and higher latitude areas.**
(DOCX)

**S10 Table. Fixed-effects model estimation of CCI and AVE through cosinor model (Model 2) with amplitude and acrophase for the lower and higher latitude areas.**
(DOCX)

**S11 Table. Fixed-effects model estimation of CCI with length of daylight, cloud cover, precipitation, and temperature (Model 3) for the lower and higher latitude areas.**
(DOCX)

**S12 Table. Fixed-effects model estimation of AVE with length of daylight, cloud cover, precipitation, and temperature (Model 3) for the lower and higher latitude areas.**
(DOCX)

**S13 Table. Fixed-effects model to predict CCI and AVE from season and latitude (Model 4) for the lower and higher latitude areas.**
(DOCX)

## Author Contributions

**Conceptualization:** Yoichi Sekizawa, Yoko Konishi.

**Formal analysis:** Yoichi Sekizawa, Yoko Konishi.

**Investigation:** Yoko Konishi.

**Methodology:** Yoichi Sekizawa, Yoko Konishi.

**Supervision:** Yoko Konishi.

**Writing – original draft:** Yoichi Sekizawa.

**Writing – review & editing:** Yoko Konishi.

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
