## [Decision Letter · Decision Letter 0]

2 Nov 2020

PONE-D-20-27021

Are consumer confidence and asset value expectations affected by length of daylight in line with the seasonal affective disorder?

PLOS ONE

Dear Dr. Sekizawa,

Thank you for submitting your manuscript to PLOS ONE. After careful consideration, we feel that it has merit but does not fully meet PLOS ONE’s publication criteria as it currently stands. Therefore, we invite you to submit a revised version of the manuscript that addresses the points raised during the review process.

In the revised version of the paper please make a clear distinction between the SAD and the other persons experiencing mild changes related to the same phenomenon. Please highlight the question you have considered in order to identify the SAD persons. If this is not the case, please properly adjust the title and the abstract in order to better reflect the type of persons under investigation. Also, additional robustness checking  is needed in order to support the results and the advancements made by the study should be discussed more in depth in comparison with other studies in the field. 

If the reviewers have suggested references to be included please do so only if they comply with the theme of the paper, otherwise, please feel free to disregard them.

We look forward to receiving your revised manuscript.

Kind regards,

Camelia Delcea

Academic Editor

PLOS ONE

Journal Requirements:

2. Please modify the title to ensure that it is meeting PLOS’ guidelines (https://journals.plos.org/plosone/s/submission-guidelines#loc-title). In particular, the title should be "specific, descriptive, concise, and comprehensible to readers outside the field" and in this case it is not informative and specific about your study's scope and methodology.

Reviewers' comments:

Reviewer's Responses to Questions

**Comments to the Author**

1. Is the manuscript technically sound, and do the data support the conclusions?

Reviewer #1: Partly

Reviewer #2: Yes

2. Has the statistical analysis been performed appropriately and rigorously? 

Reviewer #1: Yes

Reviewer #2: Yes

3. Have the authors made all data underlying the findings in their manuscript fully available?

Reviewer #1: No

Reviewer #2: Yes

4. Is the manuscript presented in an intelligible fashion and written in standard English?

Reviewer #1: Yes

Reviewer #2: Yes

5. Review Comments to the Author

Reviewer #1: The current manuscript describes a study where questionnaire answers of consumer confidence in economy are correlated to the length of daylight in Japan. Idea behind the study is that reduction in daylight would cause lower mood that would translate into more risk aversive economic behaviour and therefore would lower stock prizes.

This is an interesting idea but at the moment the manuscript has large conceptual problem and some limitations in reporting of the results. These should be addressed before it can be considered for publication.

Association between economic activity and seasons and hypothesis that this association is caused by seasonal affective disorders (SAD) are interesting. However, conceptually this study does not investigate SAD. SAD is collection of symptoms that begin with seasonal change (usually beginning of winter) and remit with new seasonal shift (usually at spring) and include changes in mood, energy level, appetite and sleep. It is often diagnosed with Seasonal Pattern Assessment Questionnaire or clinical interview. While many people experience mild changes in mood in reduced daylight only small part of them experience severe enough symptoms to qualify for SAD.

The study in question investigates whether peoples confidence in economy as measured by CCI and AVE are correlated with amount of daylight measured by date, latitude and weather. This is interesting information but it does not tell us whether the causal mechanism at play is SAD or even mood in more general terms as these are not measured. The authors acknowledge this limitation in some parts of the manuscript but title of the manuscript as well as abstract highlight investigation of SAD which is not accomplished in this study. Therefore the title and abstract should be changed to better reflect what is done in the study. Also, other possible explanations than SAD for the association could be considered.

The strength of the study is interesting sample with large N and measures of weather in addition to date and latitude to represent daylight. However more information about the CCS is needed: How were the households sampled into CCS? What, if any, descriptives, such as household size or income, are available for households? Are there studies about psychometric properties of CCI and AVE?

The regression analyses are covered in great detail in the methods. Maybe some of the formulas could be moved into supplementary material?

In Table 1. “Observation” should be changed to n.

Is it necessary to report 4 decimals in tables?

R-squared in models 1, 2 and 4 appears to be very low meaning that these models do not explain much variation of CCI and AVE. In model 3 the R-square is surprisingly high. With such a small amount of variation explained by the models 1,2 and 4, practical significance of these findings should be discussed. Also, is there logical explanation why model 3 has such a different amount of variation explained?

Reviewer #2: Introduction should emphasize more the contribution of this study related to the existing ones. It should be restructured so that in the newer version a true introduction is found (problem statement, current state in literature briefly, and contributions here), then the second section shoud be the theroy and possible causes of the SAD effects, with a third (or it could be second) section of previous related research regarding this topis. This is the most weak part of the study - previous literature overview should be included, with a critique. Some papers are missing, such as:

- https://www.mdpi.com/2227-9091/6/4/140 and the references in this paper related to SAD effects on stock markets, please include them

Empirical analysis needs additional robustness checking, it is not found here (either via chaning the variables in the model, or by estimating the model for sub-samples, etc.)

Furthermore, the discussion is too brief, it should be contrasted to related empirical studies, not just the theory.

Conclusion is too short as well, it should be a true conclusion, and not the summary (abstract) of the paper, as you already have it. It should discuss the findings in terms of policy recommendations, and recommendations for investors, the advantages and disadvantages of this study, future research suggestions.

6. PLOS authors have the option to publish the peer review history of their article (what does this mean?). If published, this will include your full peer review and any attached files.

Reviewer #1: No

Reviewer #2: No

---

## [Author Response · Author response to Decision Letter 0]

15 Dec 2020

RESPONSE TO ACADMIC EDITOR: 

Comment 1: In the revised version of the paper please make a clear distinction between the SAD and the other persons experiencing mild changes related to the same phenomenon. Please highlight the question you have considered in order to identify the SAD persons. If this is not the case, please properly adjust the title and the abstract in order to better reflect the type of persons under investigation. Also, additional robustness checking is needed in order to support the results and the advancements made by the study should be discussed more in depth in comparison with other studies in the field.

Response: We wish to thank Academic Editor for considering our paper and giving us a precious opportunity to receive comments from the reviewers, which have helped us to significantly improve our paper. As Reviewer #1 correctly pointed out, this study does not investigate SAD itself. What we did in the present study is to explore the association between the length of daylight, consumer confidence index (CCI) and asset value expectation (AVE). In order to reflect what we really did in the paper we changed the title and abstract. We also changed our introduction and discussion accordingly. 

We carried out two robustness checks. First, we divided the data from the 14-year and four months research period almost equally into two research periods of seven years and seven years and four months. Secondly, we split the whole samples into lower and higher latitude groups. Robustness checks are reported on p.23 (lines 283-308) and discussed on p. 27 (lines 369-382). 

In the discussion part, the advancements made by our study are now thoroughly explained and the insights gained from combining the results of the present study to those of other studies both in economics and psychology are now discussed in depth.

RESPONSE TO REVIEWER #1: 

We wish to thank Reviewer #1 for the insightful comments, which have helped us to significantly improve our paper. We respond to the comments as follows.

Comment 1: The current manuscript describes a study where questionnaire answers of consumer confidence in economy are correlated to the length of daylight in Japan. Idea behind the study is that reduction in daylight would cause lower mood that would translate into more risk aversive economic behaviour and therefore would lower stock prizes.

This is an interesting idea but at the moment the manuscript has large conceptual problem and some limitations in reporting of the results. These should be addressed before it can be considered for publication.

Association between economic activity and seasons and hypothesis that this association is caused by seasonal affective disorders (SAD) are interesting. However, conceptually this study does not investigate SAD. SAD is collection of symptoms that begin with seasonal change (usually beginning of winter) and remit with new seasonal shift (usually at spring) and include changes in mood, energy level, appetite and sleep. It is often diagnosed with Seasonal Pattern Assessment Questionnaire or clinical interview. While many people experience mild changes in mood in reduced daylight only small part of them experience severe enough symptoms to qualify for SAD.

The study in question investigates whether peoples confidence in economy as measured by CCI and AVE are correlated with amount of daylight measured by date, latitude and weather. This is interesting information but it does not tell us whether the causal mechanism at play is SAD or even mood in more general terms as these are not measured. The authors acknowledge this limitation in some parts of the manuscript but title of the manuscript as well as abstract highlight investigation of SAD which is not accomplished in this study. Therefore the title and abstract should be changed to better reflect what is done in the study. Also, other possible explanations than SAD for the association could be considered. 

Response: We thank Reviewer #1 for this pertinent comment. As the reviewer correctly pointed out, this study does not investigate SAD itself. What we did in the present study is to explore the association between daylight length and CCI and AVE. In order to reflect what we really did in the paper, we changed the title and abstract. 

Old Title

Are consumer confidence and asset value expectations affected by length of daylight in

line with the seasonal affective disorder?

New Title

Are consumer confidence and asset value expectations positively associated with length of daylight?: An exploration of psychological mediators between length of daylight and seasonal asset price transitions

Comment 2 The strength of the study is interesting sample with large N and measures of weather in addition to date and latitude to represent daylight. However more information about the CCS is needed: How were the households sampled into CCS? What, if any, descriptives, such as household size or income, are available for households? 

Response: We added sentences explaining how the households were sampled into CCS as follows.

“A three-level stratified sampling was used for the sampling procedure in the CCS. Before March 2013, the sample size included 6,720 households. Since April 2013, 8,400 households participated. The sampling procedure since April 2013 has been made public (https://www.esri.cao.go.jp/jp/stat/shouhi/shouhi_gaiyou.html#a2). At the first stage, 242 cities, towns, or villages from all 47 prefectures were selected, according to the results of the national census carried out every five years (the latest one occurred on Oct. 1st, 2015). This process was mostly random except that two cities were selected in scarcely populated prefectures, in which the calculated number was one. At the second stage, 336 local units were randomly selected. At the third stage, 25 households were randomly selected in each of the local units.” (p. 8, lines 95–104)

Regarding questions other than CCI and AVE in CCS, we touched upon them as follows.

“In addition to the questions on the CCI and AVE explained later, the CCS was composed of questions on the prediction of daily purchasing goods’ prices one year ahead, the sex of the household heads, whether they are in labor force, the types of jobs they take (agriculture, being employed, self-employed, or others), the age of the household heads, the number of household members, the number of those who are in labor force within the household, the number of unemployed and job-seeking persons within the household, the annual household income (seven categories), the main income categories (i.e., salary, business, pension, or others), the types of residence (i.e., self-owned independent housing, self-owned condominium unit, public rental housings, residence for employees, and private rental housings), whether they have housing mortgage, and the floor space of the residences.” (p. 8, lines 84-93)

Comment 3: Are there studies about psychometric properties of CCI and AVE?

Response: We worked on the relationship between depression and questions of CCI in another study and found a negative association between the two. Unfortunately, this paper is still under review and therefore cannot be cited. Instead, we cited other studies as follows.

“A recent study reported a positive association between consumer sentiment and mental stress [50]. Similarly, several studies confirmed the associations between the expectation of future economy and happiness/mental health [51-53]. These studies along with the present study suggest the existence of an emotional seasonal cycle in line with the existence of an optimism/pessimism cycle.” (p. 27, lines 364–368)

Comment 3: The regression analyses are covered in great detail in the methods. Maybe some of the formulas could be moved into supplementary material?

Response: According to this comment, we tried moving some parts of analytic strategies to supplementary material. Unfortunately, moving only some parts of the explanations of the analytic strategies were difficult. Therefore, we kept the original format. 

Comment 4: In Table 1. “Observation” should be changed to n.

Response: In line with the comment, we changed “Observation” to n in Table 1. Thank you for the comment. 

Comment 5: Is it necessary to report 4 decimals in tables?

Response: In line with the comment, we changed decimals in tables from 4 decimals to 3 decimals except Table 5. In Table 5, β_(LATITUDE*SEASON) was 0.0004, but significant (p < 0.001). This part is one of the most important parts of our analyses. Thus, we kept 4 decimals in this Table. We did the same way for our robustness checks (S7 Table and S8 Table). 

Comment 6: R-squared in models 1, 2 and 4 appears to be very low meaning that these models do not explain much variation of CCI and AVE. In model 3 the R-square is surprisingly high. With such a small amount of variation explained by the models 1,2 and 4, practical significance of these findings should be discussed. Also, is there logical explanation why model 3 has such a different amount of variation explained?

Response: We have revised the text on the basis of the comment as follows:

“There are some limitations in the present study. First, the adjusted R-squared for the models, except Model 3, are below 0.01, suggesting that the explanatory power is quite low. It is high for Model 3, but this reflects the fact that the CCIs of the previous months are included as explanatory variables in the model.” (p. 28, lines 388–391)

RESPONSE TO REVIEWER #2: 

We wish to express our appreciation to the Reviewer for the insightful comments, which have helped us to significantly improve the paper.

Comment 1: Introduction should emphasize more the contribution of this study related to the existing ones. It should be restructured so that in the newer version a true introduction is found (problem statement, current state in literature briefly, and contributions here), then the second section shoud be the theroy and possible causes of the SAD effects, with a third (or it could be second) section of previous related research regarding this topis. This is the most weak part of the study - previous literature overview should be included, with a critique. 

Response: We thank the Reviewer for this pertinent comment. In line with the comment as well as the guideline of PLOS ONE, we completely revised our introduction. For example, two level 2 headings (“A brief review of the relevant literature and controversies” and “Overall Aim of the present study”) and corresponding descriptions were incorporated in the paper.

Comment 2: Some papers are missing, such as:

- https://www.mdpi.com/2227-9091/6/4/140 and the references in this paper related to SAD effects on stock markets, please include them

Response: We revised the text and touched upon stock markets and cited relevant studies (underlined part). 

“Inspired by the research conducted by Kamstra et al. [2], seasonal variations consistent with SAD have been reported in the pricing of initial public offerings [3, 5, 6], art auction prices [7], real estate prices [4], bond prices [8], as well as stock prices in several countries [9, 10, 13].” (p. 5, lines 36-39)

9. Sakakibara S, Yamasaki T, Okada K. The Calendar Structure of the Japanese Stock Market: The ‘Sell in May Effect’ Versus the ‘Dekansho-Bushi Effect’. In: Ikeda S, Kato HK, Ohtake F, Tsutsui Y, editors. Behavioral Interactions, Markets, and Economic Dynamics: Topics in Behavioral Economics. Tokyo: Springer Japan; 2016. p. 637-61.

10. Škrinjarić T. Testing for Seasonal Affective Disorder on Selected CEE and SEE Stock Markets. Risks. 2018;6(4):140.

13. Murgea A. Seasonal affective disorder and the Romanian stock market. Economic Research-Ekonomska Istraživanja. 2016;29(1):177-92. doi: 10.1080/1331677X.2016.1164924.

Comment 3: Empirical analysis needs additional robustness checking, it is not found here (either via chaning the variables in the model, or by estimating the model for sub-samples, etc.)

Response: We thank the Reviewer for this pertinent comment. We carried out two robustness checks. First, we divided the data from the 14-year and four months research period almost equally into two research periods of seven years and seven years and four months. Secondly, we split the whole samples into lower and higher latitude groups. Their results are explained as follows. 

“We performed two robustness checks. First, we divided the whole periods into two. The data we used in the present study is from April 2004 to August 2018, for a total of 173 months. The first period was from April 2004 to March 2011 (84 months) and the second was from April 2011 to August 2018 (89 months). The monthly summary statistics of CCI and AVE for the two periods are shown in S2 Table. The results of Model 1 are shown in S3 Table and graphically shown in S1 Fig. The results of Model 2 are shown in S4 Table and graphically shown in S2 Fig. The results of Model 3 are shown in S5 Table for CCI and S6 Table for AVE. The results of Model 4 are shown in S7 Table. In general, the seasonal pattern is less clear and smaller in the latter period, as shown in S1 Fig and S2 Fig. However, most key coefficients (such as β_DAYLIGHT in Model 3 (S5 Table) and β_(LATITUDE*SEASON) in Model 4 (S7 Table)) remained significant except β_(LATITUDE*SEASON) in AVE until March 2011 (S7 Table). 

In the second robustness check, we divided the samples into two according to the latitude by the median. The total number of participants was 85,730. The latitude of the first group was below 35.41 and the latitude of the second group was above 35.41. In the lower latitude areas, there were 43,289 participants and the mean latitude for them was 34.133. In the higher latitude areas, there were 42,464 participants and the mean latitude for them was 37.060. The monthly summary statistics of CCI and AVE for the two groups are shown in S8 Table. The results of Model 1 are shown in S9 Table and graphically shown in S3 Fig. The results of Model 2 are shown in S10 Table and graphically shown in S4 Fig. The results of Model 3 are shown in S11 Table for CCI and S12 Table for AVE. The results of Model 4 are shown in S13 Table. In general, the seasonal pattern remains stable regardless of the latitude (S3 Fig and S4 Fig). β_DAYLIGHT in Model 3 (S11 Table and S12 Table) were significant. β_(LATITUDE*SEASON) in Model 4 for both CCI and AVE in higher latitude areas were not significant (S13 Table).” (p. 23, line 283-308)

We also discussed robustness check in the Discussion as follows.

“The results from the robustness checks should be interpreted cautiously. When the data were split by time, the seasonal patterns were less clear and smaller in the latter period compared to the former one. One explanation is that CCI and AVE are moved by various events in the short term, whereas the effects of the daylight length is rather stable and easier to observe in the long-term data, in which the effects of various short-term events are cancelled out. By contrast, dividing the data by the latitude showed similar patterns for both lower and higher latitude areas. If CCI and AVE were affected by the length of daylight, differences of CCI and AVE between winter and summer should be larger in the higher latitude areas than the lower latitude areas. Indeed, the amplitude of CCI and AVE were larger in the higher latitude area group than the lower latitude area group, but the difference was not significant (S10 Table and S4 Fig). One possible explanation is that the difference in the latitude between the two groups is too small to show any significant effect. The mean latitude was 34.133 for the lower latitude group and 37.060 for the higher latitude group. This difference of around 300 km may be too small to find the difference. Studies in other countries are required to further explore this.” (p. 27, lines 369-382)

Comment 4: Furthermore, the discussion is too brief, it should be contrasted to related empirical studies, not just the theory.

Response: We thank the Reviewer for this pertinent comment.

In line with the comment from the reviewer we wholly rewrote the discussion part (p. 24, lines 309 -403). We accommodated related empirical studies. The original submission contained 556 words in the discussion part, but the revised version has 1330 words. 

Comment 5: Conclusion is too short as well, it should be a true conclusion, and not the summary (abstract) of the paper, as you already have it. It should discuss the findings in terms of policy recommendations, and recommendations for investors, the advantages and disadvantages of this study, future research suggestions.

Response: We rewrote the conclusion accommodating the reviewer’s comment. Unfortunately, it is too early to draw policy implication from the present study only. Thus, we wrote about the expectations on future research including future research suggestions as follows (underlined part).

“Analyzing the individual longitudinal data of CCI and AVE generated from raw data between 2004 and 2018, we found a seasonal cycle. The trough of this cycle occurred in December and the peak in the early summer. We also found that the higher the latitude, the larger the seasonal cycle of CCI and AVE. These results suggest that differences in the length of daylight may cause asset price transitions through the seasonal optimism/pessimism cycle as shown by the CCI and AVE. To further explore the causal relationship between these variables and draw reliable policy implications, the construction of individual longitudinal data merging asset purchasing behavior data and consumer confidence data is needed.” (p. 29, lines 405-413)

CORRECTION FROM AUTHORS

We found two mistakes to be corrected in our original manuscript. We apologize for our mistakes. We corrected them as follows.

First Correction: 

The underlined part on p. 16 (lines 233-234) was corrected as follows.

(Original submission)

Table 1 shows the monthly data for CCI and AVE from April 2004 until August 2018, with the highest values occurring in June and May, respectively.

(Revised submission)

Table 1 shows the monthly data for CCI and AVE from April 2004 until August 2018, with the highest values occurring in May.

Second Correction: 

AVE parts in Table 4 in the original submission was incorrect. We mistakenly used lagged CCI instead of lagged AVE for independent variables. We corrected them in the resubmitted version. (p. 21).

---

## [Decision Letter · Decision Letter 1]

2 Jan 2021

Are consumer confidence and asset value expectations positively associated with length of daylight? : An exploration of psychological mediators between length of daylight and seasonal asset price transitions

PONE-D-20-27021R1

Dear Dr. Sekizawa,

We’re pleased to inform you that your manuscript has been judged scientifically suitable for publication and will be formally accepted for publication once it meets all outstanding technical requirements.

Kind regards,

Camelia Delcea

Academic Editor

PLOS ONE

Additional Editor Comments (optional):

Reviewers' comments:

Reviewer's Responses to Questions

**Comments to the Author**

1. If the authors have adequately addressed your comments raised in a previous round of review and you feel that this manuscript is now acceptable for publication, you may indicate that here to bypass the “Comments to the Author” section, enter your conflict of interest statement in the “Confidential to Editor” section, and submit your "Accept" recommendation.

Reviewer #2: All comments have been addressed

2. Is the manuscript technically sound, and do the data support the conclusions?

Reviewer #2: Yes

3. Has the statistical analysis been performed appropriately and rigorously? 

Reviewer #2: Yes

4. Have the authors made all data underlying the findings in their manuscript fully available?

Reviewer #2: Yes

5. Is the manuscript presented in an intelligible fashion and written in standard English?

Reviewer #2: Yes

6. Review Comments to the Author

Reviewer #2: The authors have resolved my issues. Thus it is ready to be published in my opinion. Although the title of the article is too lenghty now in my opinion, the other reviewer asked for the title to reflect the paper.

7. PLOS authors have the option to publish the peer review history of their article (what does this mean?). If published, this will include your full peer review and any attached files.

Reviewer #2: No

---

## [Editor Report · Acceptance letter]

6 Jan 2021

PONE-D-20-27021R1 

Are consumer confidence and asset value expectations positively associated with length of daylight? : An exploration of psychological mediators between length of daylight and seasonal asset price transitions 

Dear Dr. Sekizawa:

I'm pleased to inform you that your manuscript has been deemed suitable for publication in PLOS ONE. Congratulations! Your manuscript is now with our production department. 

Kind regards, 

on behalf of

Dr. Camelia Delcea 

Academic Editor

PLOS ONE